# Gene Panel Testing for Breast Cancer Reveals Differential Effect of Prior *BRCA1/2* Probability

**DOI:** 10.3390/cancers13164154

**Published:** 2021-08-18

**Authors:** D. Gareth Evans, Elke M. van Veen, Emma R. Woodward, Elaine F. Harkness, Jamie M. Ellingford, Naomi L. Bowers, Andrew J. Wallace, Sacha J. Howell, Anthony Howell, Fiona Lalloo, William G. Newman, Miriam J. Smith

**Affiliations:** 1NW Genomic Laboratory Hub, Manchester Centre for Genomic Medicine, Manchester University Hospitals NHS Foundation Trust, Manchester M13 9WL, UK; Elke.vanVeen@manchester.ac.uk (E.M.v.V.); Emma.Woodward@mft.nhs.uk (E.R.W.); jamie.ellingford@manchester.ac.uk (J.M.E.); Naomi.Bowers@mft.nhs.uk (N.L.B.); andrew.wallace@mft.nhs.uk (A.J.W.); Fiona.Lalloo@mft.nhs.uk (F.L.); William.newman@manchester.ac.uk (W.G.N.); miriam.smith@manchester.ac.uk (M.J.S.); 2Division of Evolution and Genomic Sciences, School of Biological Sciences, Faculty of Biology, Medicine and Health, University of Manchester, Manchester Academic Health Science Centre, Manchester M13 9WL, UK; 3Prevent Breast Cancer Centre, Wythenshawe Hospital Manchester Universities Foundation Trust, Wythenshawe, Manchester M23 9LT, UK; Elaine.F.Harkness@manchester.ac.uk (E.F.H.); sacha.howell@manchester.ac.uk (S.J.H.); anthony.howell@manchester.ac.uk (A.H.); 4Manchester Breast Centre, The Christie NHS Foundation Trust, Wilmslow Road, Manchester M20 4BX, UK; 5Division of Cancer Sciences, Faculty of Biology, Medicine and Health, University of Manchester, Manchester Academic Health Science Centre, Manchester M20 4GJ, UK

**Keywords:** breast cancer, *BRCA1*, *BRCA2*, *PALB2*, *ATM*, *CHEK2*, panel test

## Abstract

**Simple Summary:**

Previous studies reporting large gene panels in breast cancer have mainly reported on the contribution of additional genes compared to *BRCA1/2*. We have shown a very large differential additional effect from non-BRCA genes dependent on a priori likelihood of *BRCA1* and *BRCA2* combined. We have undertaken panel testing beyond *BRCA1/2* in 1398 patients with breast cancer and identified 95 (6.3%) with actionable breast cancer genes. The highest rate was found for grade 3 ER+ Her2− breast cancers. Those with the lowest likelihood of *BRCA1/2* by Manchester score had a 3-fold higher rate of non-BRCA genes whereas those with predicted rates of ~80% had 5-fold higher rate of *BRCA1/2.* Unless those referring patients with breast cancer for extended panel testing are certain there is no loss of sensitivity for *BRCA1/2* they should opt of a bespoke *BRCA1/2* test first in those with high prior likelihoods of *BRCA1/2* pathogenic variants.

**Abstract:**

Whilst panel testing of an extended group of genes including *BRCA1/2* is commonplace, these studies have not been subdivided by histiotype or by a priori *BRCA1/2* probability. Patients with a breast cancer diagnosis undergoing extended panel testing were assessed for frequency of actionable variants in breast cancer genes other than *BRCA1/2* by histiotype and Manchester score (MS) to reflect a priori *BRCA1/2* likelihood. Rates were adjusted by prior testing for *BRCA1/2* in an extended series. 95/1398 (6.3%) who underwent panel testing were found to be positive for actionable non-*BRCA1/2* breast/ovarian cancer genes (*ATM*, *BARD1*, *CDH1*, *CHEK2*, *PALB2*, *PTEN*, *RAD51C*, *RAD51D*, *TP53*). As expected, *PALB2*, *CHEK2* and *ATM* were predominant with 80-(5.3%). The highest rate occurred in Grade-3 ER+/HER2− breast cancers-(9.6%). Rates of non-BRCA actionable genes was fairly constant over all likelihoods of *BRCA1/2* but adjusted rates were three times higher with MS < 9 (*BRCA1/2* = 1.5%, other = 4.7%), but was only 1.6% compared to 79.3% with MS ≥ 40. Although rates of detection of non-BRCA actionable genes are relatively constant across *BRCA1/2* likelihoods this disguises an overall adjusted low frequency in high-likelihood families which have been heavily pre-tested for *BRCA1/2*. Any loss of detection sensitivity for *BRCA1/2* actionable variants in breast cancer panels should lead to bespoke *BRCA1/2* testing being conducted first.

## 1. Introduction

In recent years there has been a massive expansion in the use of multi-gene panels to test for breast cancer predisposition. These results have been used as a type of case control study to assess genes for breast cancer associations and the increase in actionable gene identification other than *BRCA1* and *BRCA2* [1,2,3,4,5,6,7,8,9,10]. These studies have generally shown an almost equal rate of pathogenic variants (PVs) identified in *BRCA1/2* compared to all of the additional potentially actionable genes combined. In addition, many of the additional variants have been identified in the lower penetrance moderate risk genes [11], *ATM* and *CHEK2*, that occur with a higher population prevalence. The only other actionable breast cancer gene consistently identified at a substantial rate is *PALB2*, which is now considered to be a high-risk gene [12]. Many studies have also included a range of organ sites, as well as including unaffected individuals [2,7,8,9].

In general, these studies have concentrated on the headline rates of pathogenic variants (PVs) identified in the additional genes and not on the differential effect of the original likelihood of a *BRCA1/2* pathogenic variant. Most studies that have used a ‘control’ frequency for PVs in the relevant genes have used population databases such as gnomAD [13], which may not be ideal controls for the population tested as they may not represent the rates in certain populations especially those with founder effects. We have assessed the frequency of potentially actionable breast cancer gene variants in a large series of women with breast cancer comparing rates of detection of non-*BRCA* PVs with *BRCA1/2* at different *BRCA1/2* likelihood scores. We have also assessed the variant frequencies in different breast pathology subtypes.

## 2. Materials and Methods

### 2.1. Study Population

A total of 5060 women with breast cancer and 1443 control samples were included in the study, male cases were excluded. Of the patients with a breast cancer diagnosis, 4536 were referred to St Mary’s, Wythenshawe and the Christie hospitals in Manchester, consented and had blood samples taken for DNA extraction testing of breast cancer genes through the Manchester Centre for Genomic Medicine (MCGM) (Figure 1). MCGM participants agreed to return of results on genes predisposing to breast cancer. The remaining participants were recruited through the Predicting the Risk Of Cancer At Screening (PROCAS) study. This study recruited women aged 46–73 years of age from a population screening programme and included 187 breast cancers diagnosed before study entry and 337 after study entry (total *n* = 524). In addition, 1443 control samples without breast cancer from PROCAS [14] were tested as part of Breast Cancer Risk after Diagnostic Gene Sequencing (BRIDGES). PROCAS women only initially provided consent for return of a polygenic risk score, but also consented to further research genetic testing. We are applying for ethical approval to approach women with panel testing to assess whether they wish to receive results.

### 2.2. Genetic Testing

Clinical pre-screening for *BRCA1/2* variants was carried out on 4142 MCGM patients. Samples from 874 women were selected for Next Generation Sequencing (NGS) panels, including 480 who tested negative on *BRCA1/2* pre-screening and 394 who had not undergone pre-screening. The panel included as a minimum the following genes in addition to *BRCA1* and *BRCA2*: *PALB2*, *CHEK2*, *ATM*, *CDH1*, *STK11*, *PTEN*, *TP53*, *RAD51C*, *RAD51D*, *BRIP1*, *BARD1* and *NBN*. Saliva DNA samples from the PROCAS study [15] were tested as part of the BRIDGES study utilising a 34 gene panel (*ABRAXAS1*, *AKT1*, *ATM*, *BABAM2*, *BARD1*, *BRCA1*, *BRCA2*, *BRIP1*, *CDH1*, *CHEK2*, *EPCAM*, *FANCC*, *FANCM*, *GEN1*, *MEN1*, *MLH1*, *MRE11*, *MSH2*, *MSH6*, *MUTYH*, *NBN*, *NF1*, *PALB2*, *PIK3CA*, *PMS2*, *PTEN*, *RAD50*, *RAD51C*, *RAD51D*, *RECQL*, *RINT1*, *STK11*, *TP53*, *XRCC2*) including the four mismatch repair (MMR) genes (*MLH1*, *MSH2*, *MSH6* and *PMS2*) [16].

PVs were annotated according to the ACMG guidelines [17] and verified using Sanger sequencing. Additionally, the PV rate of all 5060 women with breast cancer tested for *BRCA1/2* at the MCGM (1997–2021) and in PROCAS was also assessed, regardless of whether they have had (additional) panel testing. The samples clinically tested at MCGM were annotated by a clinically approved national laboratory hub. Annotation of research results from BRIDGES were initially confirmed by the BRIDGES team and confirmed by our in-house research group (EvV).

### 2.3. Calculated Likelihood of a BRCA1/1 Pathogenic Variant

The pathology adjusted Manchester Scoring System (MS) was used to assess likelihoods of a pathogenic variant in *BRCA1/2* [18]. Briefly, each breast cancer in a direct lineage to the index case is scored based on age at diagnosis for each gene (1–6 points) as well as each non mucinous epithelial ovarian cancer (5–8 points). All breast and ovarian cancers in the index individual are scored with an adjustment for pathology. Her2+, low grade and lobular cancers receive minus scores for *BRCA1* whilst triple negative and high-grade cancers receive positive scores. Pathology adjusted scores of 15–19 are equivalent to a 10% probability of a *BRCA1/2* pathogenic variant with a 20–24-point score equivalent to a 20% probability [18]. The proportion of positive tests from panel testing other than *BRCA1/2* was adjusted to reflect previous testing of *BRCA1/2* and the proportion of positive tests for *BRCA1/2* at each MS score range. Thus a 10% detection rate in *BRCA1/2* negative samples where the *BRCA1/2* detection was 50% translates to a 5% overall panel detection rate beyond BRCA. An actionable breast cancer gene was defined as a gene confirmed in the BRIDGES study as having a 2-fold relative risk for breast cancer [16].

## 3. Results

### 3.1. Participants

A total of 1398 people with breast cancer were included in this study. There were 874 women tested through the MCGM as part of the clinical service or research projects. Of these, 740 women had a family history of only breast cancer, with 134 having an additional personal or family history of ovarian cancer. Additionally, 524 affected women and 1443 female controls that took part in the PROCAS study were included (age range 29–75; median 58.8 years) and were tested as part of the BRIDGES study (Figure 1).

Overall, 480 women with a family history (34.3%) and MS of −1 (some score below 0 with pathology adjustment) up to 56 (median 18) had been negative for *BRCA1/2* on a pre-screen.

All panels included the following genes: *PALB2*, *CHEK2*, *ATM*, *CDH1*, *STK11*, *PTEN*, *TP53*, *RAD51C*, *RAD51D*, *BRIP1*, *BARD1* and *NBN*. All 524 population-based samples and 347 familial samples (total *n* = 871) were also tested for variants in mismatch repair (MMR) genes. In addition, 1443 controls without breast cancer from PROCAS aged 46–73 years were tested for the full panel, including MMR genes.

Additionally, the PV status of all 5060 women with breast cancer tested for *BRCA1/2* was included in order to determine the effect of extended panel testing in women stratified by MS.

### 3.2. Pathogenic Variant Rate

Overall, 174/1398 (12.4%) PVs were identified in 172 women (*BRCA1* (26); *BRCA2* (43); *ATM* (31); *BARD1* (1); *BRIP1* (7); *CDH1* (1); *CHEK2* (25); *NBN* (3); *PALB2* (24); *PTEN* (1); *RAD51C* (1); *RAD51D* (1); *TP53* (10)) (Table 1). One woman harboured both a *BRCA1* and a *BRCA2* PV and one woman carried a *BRCA2* PV as well as a *BRIP1* PV.

Of the 874 women tested via MCGM, 60 *BRCA1*/*2* PVs were identified in 59 women (*BRCA1* (25); *BRCA2* (35) (one harboured both a *BRCA1* and a *BRCA2* PV), however 480 samples had already screened negative for *BRCA1/2.* This thus represents a detection rate of 60/394 (15%). In the remaining 815 women, 80 PVs were identified in non-*BRCA1/2* genes (*ATM* (25); *BARD1* (1); *BRIP1* (5); *CDH1* (1); *CHEK2* (15); *NBN* (2); *PALB2* (20); *RAD51C* (1); *RAD51D* (1); *TP53* (9)).

Of the 524 women recruited through PROCAS, 34 PVs in 33 women were identified (*BRCA2* (8); *BRCA1* (1); *ATM* (6); *BRIP1* (2); *CHEK2* (10); *NBN* (1); *PALB2* (4); *PTEN* (1); *TP53* (1); one woman harboured both a *BRCA2* and a *BRIP1* PV).

A total of 95 of 1398 (6.3%) women who underwent panel testing were found to be positive for actionable non-*BRCA1/2* breast/ovarian cancer genes (*ATM*, *BARD1*, *CDH1*, *CHEK2*, *PALB2*, *PTEN*, *RAD51C*, *RAD51D*, *TP53*). As expected, *PALB2*, *CHEK2* and *ATM* were the most frequent with 80 (5.3%) PVs accounting for 84.2% non *BRCA1/2* actionable results. Of the other high-risk genes there were 10 *TP53*, one *CDH1* and one *PTEN* PVs, but none in *STK11*. All the expected breast cancer associated genes (*ATM*, *CHEK2*, *PALB2*, *TP53*) had significantly increased odds ratios of above 2-fold compared to the 1443 PROCAS control samples (Table 1). Other PVs were identified in genes that have a less clear association with breast cancer such as: *BRIP1*, *NBN*, *RAD50* and *RECQL*, but none of these showed significant associations.

### 3.3. Detection Rate by Manchester Score

The detection rates of PVs in non-*BRCA1/2* genes in the full extended panel cohort did not vary substantially by MS (ranging between 5.3–11.4%). However, when controlling for those with prior *BRCA1/2* testing, the added value of the extended panel was actually lower in those with a higher MS (i.e., those individuals with a higher probability of a *BRCA1/2* pathogenic variant). Only 2.97% of those with MS ≥ 30 tested positive for a non-*BRCA1/2* PV when allowing for the higher likelihood of a *BRCA1/2* positive result of 63.2% (251/397) in the group of all individuals tested for *BRCA* PVs.

There was no significant difference between rates of the most commonly detected non-*BRCA* PVs in those with MS ≥ 15 and <15 (*ATM* PVs (3.0% and 1.6% *p* = 0.098), *CHEK2* PVs (1.7% and 1.9% *p* = 0.84) and *PALB2* PVs (2.5% and 1.1% *p* = 0.0605)). Although the combined total of all PVs in all three genes with MS ≥ 15 was statistically significantly higher at 7.2% (43/598) vs. 4.6% (37/800)-*p* = 0.05 this did not take into account pre-screening for *BRCA1/2*.

Of all women tested for *BRCA1/2* PVs (*n* = 5060), 764 (15.10%) harboured a PV in *BRCA1/BRCA2* (Table 1). In this group, 92/116 (79.3%) women with breast cancer and a MS of ≥40 tested positive for a *BRCA1*/2 PV (*BRCA1* (*n* = 67) or *BRCA2* (*n* = 25)). In the study cohort, only an additional two non-*BRCA* PVs were identified in the group of women with a MS ≥ 40 (2/15 (13.3%) and a negative pre-screen). The adjusted proportion of PVs in non-BRCA genes of 75/1204 (6.2%) for MS ≤ 10 was significantly higher than for those with scores ≥30 (12/405-3.0%-*p* = 0.01).

Of the 918 women in the study cohort without pre-screening, 68 (7.4%) tested positive for PVs in *BRCA1* (*n* = 25), *BRCA2* (*n* = 42) or both (*n* = 1). There was an increase in *BRCA1/2* PV detection rate with increasing MS, similar to the 5060 women tested only for *BRCA1/2* PVs.

Of the 871 women tested for variants in the MMR genes, very few PVs were identified. There were only two cases with an *MSH6* PV (Appendix A). No PVs in *MLH1* or *MSH2* were identified. Similarly, in the control group there was only one *MSH6* and one *MSH2* PV identified.

Only 9/524 (1.7%) population-based PROCAS samples tested positive for a *BRCA1/2* PV (MS: 0–33 median = 3, *BRCA1* (*n* = 1), *BRCA2* (*n* = 8)). Only 28/524 samples met the MS ≥ 15 score (indicating a 10% threshold was met) and only 2/28 (7.1%) had a *BRCA1/2* PV (*BRCA1* = 1, *BRCA2* = 1). However, two women in this group had a *PALB2* PV. In those who did not meet the 10% threshold (MS < 15), there were 7/496 (1.4%) *BRCA1/2* PVs detected. In familial samples, 37/124 (29.8%) with MS ≥ 20, 11/91 (12.1%) with MS 15–19 and 5/173 (2.9%) with MS < 15 testing positive for *BRCA1/2*.

A total of 134 women with breast cancer had either a personal (*n* = 25) or family history (*n* = 109) of ovarian cancer. Detection rates were similar to those with breast cancer only with 8/134 (6%) having a PV beyond *BRCA1/2 (PALB2* = 2) and 2/25 (8%) with breast and ovarian double primaries (Appendix A).

### 3.4. Detection Rate by Tumour Pathology

Breast cancer cases with tumour pathology reports, including human epidermal growth factor receptor 2 (HER2) status are presented in Table 2. The lowest overall detection rate (6.1%). for any PV, including *BRCA1/2*, was in grade 1 estrogen receptor (ER)+/HER2− breast cancers. The highest detection rates were in grade 3 ER+/HER2− (18.7%) and triple negative cases (17.5%).

## 4. Discussion

The present study has confirmed some utility for breast cancer gene panels extended beyond *BRCA1/2* in women with breast cancer. Nonetheless, the increased diagnostic yield is dominated by just three additional genes (*PALB2*, *ATM* and *CHEK2*) accounting for over 80% of non-*BRCA1/2* variants found. Additionally, although *BRCA1/2* pre-testing increases the rate of pathogenic variant detection in these genes, with higher familial scores represented by the MS, the added benefit from a panel is lower when adjusting for a higher likelihood of *BRCA1* and *BRCA2* PVs. For instance, an index case with a MS ≥ 40 would have an ~80% chance of a *BRCA1/2* PV, but only a 1–2% chance of a PV in another actionable gene. In contrast, with a MS ≤ 8 there is only around a 1% chance of a *BRCA1/2* PV, but a 5% chance of finding another PV (mainly in *ATM* or *CHEK2*). Previous studies have not addressed this large differential effect. For those with a very high likelihood of a *BRCA1/2* PV by an accepted method (MS, BOADICEA/CANRISK, BRCAPRO), any loss in sensitivity of identification of a *BRCA1/2* PV would mitigate any additional benefit from an extended panel vs. a targeted *BRCA1/2* approach. Some loss of sensitivity is accepted in most panel tests but even a drop by 2% for a family with a MS ≥ 40 would be sufficient to warrant bespoke *BRCA1/2* testing as the first test. Although most NGS panels claim to identify large single or multiple exon copy number variations (CNVs), the proven sensitivity of these tests is less clear. In Manchester, 20% of non-Jewish *BRCA1* PVs are CNVs that in the past have been detected by multiple ligation-dependent probe amplification (MLPA). Given that any additional gene identified is likely to have lower penetrance than *BRCA1* or *BRCA2* e.g., in a large family with multiple breast/ovarian cancer cases (MS ≥ 40), it is unlikely that this will be the entire explanation for the pattern observed. For instance we observed a *RAD51C PV* in a large breast/ovary family which probably only fully explains the ovarian cancer predisposition [19], and for the *ATM* and *CHEK2* variants in six families with MS of 30–39 and one *CHEK2* in a MS of >40, these are unlikely to provide a full explanation for the pattern of disease in these families (Table 1) [11]. We have previously shown that a pathogenic *RAD51D* variant in a family with three women with ovarian cancer and four with breast cancer(MS = 55) fully segregated with disease and that *BRCA1/2* testing, including RNA analysis was negative [19]. Given that only a two-fold increased likelihood of breast cancer is associated with *RAD51D* PVs, it is still unlikely that *RAD51D* fully explains the breast cancer risk in this family, although polygenic risk could add somewhat to the effects of *RAD51D*. Although results from BRIDGES [16] and other studies clearly differentiate *PALB2* as a high-risk gene and *CHEK2* as moderate-risk, this is not reflected in the similar ORs for *CHEK2* and *PALB2* of around eight-fold, but these risk estimates have large confidence intervals due to the small number of controls with identified variants and are based on testing many familial families with pre-screens of *BRCA1/2*.

Case control analysis is arguably the most informative method to identify breast cancer gene associations as it also provides confirmation of the level of any increased risk. Many recent large-scale studies have published regarding this, including recent evidence that variants in the MMR genes *MSH6* and *PMS2* may be associated with an increased risk of breast cancer [2,20]. Although the first of these studies was a true case control study, there has been concern that one of the variant calls in *MSH6* (c.2945delC; p.(Pro982Leufs)) that was very frequent in cases, may be spurious due to a sequencing artifact [21]. This study also used gnomAD controls [13] rather than a matched, local population control series. The second study was not a true case control study and assessed risk in carriers identified on panel testing without adjusting for the proportion of referred cases with breast cancer [20]. The very high population frequencies of variants in *MSH6* and *PMS2* without typical Lynch syndrome associated cancers may ironically have falsely elevated the odds ratio [22]. Furthermore, no evidence for any association between these variants in MMR genes and breast cancer has been seen in larger case control studies [1], segregation analysis [23], nor in prospective analysis of PV carriers [24], with the possible exception of *MSH6* in BRIDGES, which was borderline significant at two-fold risk [16]. As such, patients with MMR PVs, with the possible exception of *MSH6*, should not be advised that they are at increased risk of breast cancer.

The lowest detection rate overall both for *BRCA1/2* and additional panel genes was for grade 1 cancers with 4.1% and 4.5% respectively. The highest detection rate was for grade 3 ER+ HER2− breast cancers with a non *BRCA1/2* panel detection rate of 10%. There has been very little in the literature on the effect of pathology on panel detection rates in non-*BRCA1/2* genes although we have recently reported an association with *PALB2* and grade 3 [25]. A Brazilian study recently also confirmed a higher detection rate in high grade tumours [26]. Although this was a small study (*n* = 224; 61 PVs-19 *BRCA1/2*) and the association was likely driven primarily by *BRCA1/2.* The BRIDGES study did not assess grade, but there was a clear association between *RAD51C*, *RAD51D*, *PALB2* and *BARD1*, with ER− breast cancer and ATM and CHEK2 with ER+ breast cancer with no association of ATM with ER− [16]. 

The present study does have some limitations. The size of the control population was probably not large enough to generate very reliable odds ratio estimates. We had to use our previous population estimate for *TP53* [27], but had no such estimate for other rare genes such as *CDH1*. The study was not large enough to statistically assert that *BRIP1* is not an actionable breast cancer gene [28]. Nor was it large enough to address the variant detection rates for the ovarian susceptibility genes *RAD51C* and *RAD51D* that have previously been excluded as breast cancer susceptibility genes [29,30], but which were more recently confirmed as breast cancer genes just reaching two-fold relative risk [16]. There is also some previous evidence that variants in *RAD51D* may predispose specifically to triple negative breast cancer [16]. The BRIDGES study also only called definitive PVs using NGS and did not call CNVs. As already stated, this could have reduced PV detection sensitivity in the UK population to below 80% for *BRCA1*. The odds ratios for *BRCA1* could therefore be exaggerated by the fact that in-house testing of over 4536 cases in Table 1 included MLPA, since CNVs were not reported in the control samples tested in the BRIDGES study. Despite this, our study also has a number of strengths. This is a large study with well documented family history, including first and second-degree relatives. A high proportion of the study patients had full pathology available, including HER2 status, which was not generally available before 2000. We have also used matching local controls known not to have breast cancer. 

## 5. Conclusions

Although the trend towards panel testing beyond *BRCA1/2* is likely to be irreversible, there are some lessons to be learnt from the current study. The majority of the increased diagnostic yield results from just three genes *PALB2*, *CHEK2* and *ATM.* Only *PALB2* can be considered a strong enough risk factor on its own and the actionability of *ATM* and *CHEK2* pathogenic variants needs to be taken in the context of other risk factors, including a polygenic risk score, ideally through a validated model such as BOADICEA/CanRisk [31]. Women with certain tumours, and in particular grade 1 invasive breast cancers, should be informed that there is a very low likelihood of any meaningful result from panel testing. Finally, the low adjusted diagnostic uplift in those with very high *a priori* likelihood of *BRCA1/2* means that clinicians should ensure that panels used have high sensitivity for *BRCA1/2* PVs present in their population or opt for bespoke *BRCA1/2* testing in the first instance.

## Figures and Tables

**Figure 1 cancers-13-04154-f001:**
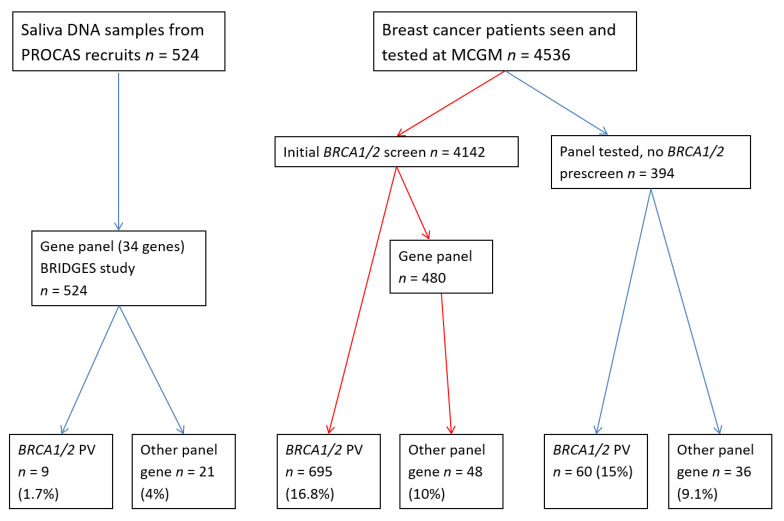
Flow chart showing selection of samples for panel testing.

**Table 1 cancers-13-04154-t001:** Rates of detection of BRCA1/2 and non BRCA breast/ovarian pathogenic variants by Manchester score.

Manchester Score	≤8	9–10	11–12	13–14	15–19	20–24	25–29	30–39	≥40	Total PVs Cases	PROCAS Controls PVs	OR	95% CI	*p*-Value
Full extended panel cohort														
*BRCA1*	1	0	0	0	2	3	3	9	8	26	2	NA	NA	NA
*BRCA2*	5	1	1	4	11	4	9	7	1	43	6	NA	NA	NA
*ATM*	7	2	1	3	7	5	3	3	0	31	5	6.52	2.56–15.55	<0.0001
*CHEK2*	12	0	1	2	4	2	0	3	1	25	3	8.74	2.88–27.60	<0.0001
*TP53 **	3	0	0	0	4	1	1	0	1	10	0	36.02	6.33–392.1	<0.0001
*PALB2*	2	3	3	1	6	6	3	0	0	24	3	8.38	2.74–26.53	<0.0001
*NBN*	1	1	0	0	1	0	0	0	0	3	2	1.55	0.32–8.75	0.6825
*CDH1*	0	0	0	1	0	0	0	0	0	1	0	n/a		
*RAD51C*	0	0	0	0	0	0	1	0	0	1	0	n/a		
*RAD51D*	0	0	0	0	0	0	1	0	0	1	0	n/a		
*BRIP1*	1	0	0	2	1	3	0	0	0	7	2	3.63	0.82–17.31	0.1037
*BARD1*	1	0	0	0	0	0	0	0	0	1	1	1.03	0.05–19.62	>0.9999
*PTEN*	1	0	0	0	0	0	0	0	0	1	0	n/a		
Total PVs	34	7	6	13	36	24	21	22	11	174	24			
Total tested	525	78	91	106	255	149	95	72	27	1398	1443			
% PVs	6.5%	9.0%	6.6%	12.3%	14.1%	16.1%	22.1%	30.6%	40.7%	12.4%	1.7%			
Total non-BRCA PVs	28	6	5	9	23	17	9	6	2	105				
% non-BRCA PVs	5.3%	7.7%	5.5%	8.5%	9.0%	11.4%	9.5%	8.3%	7.4%	7.5%				
% adjusted for all *BRCA1/2* tested	5.3%	7.4%	5.3%	8.0%	8.1%	8.8%	6.4%	3.6%	1.5%	6.4%				
No pre-screen														
*BRCA1*	1	0	0	0	2	3	3	9	8	26	2	21.00	5.41–90.02	<0.0001
*BRCA2*	5	1	1	4	11	4	9	7	1	43	6	11.77	5.05–25.65	<0.0001
*ATM*	6	2	0	3	3	3	1	1	0	19	5	6.08	2.43–14.96	<0.0001
*CHEK2*	12	0	0	0	0	0	0	1	0	13	3	6.89	2.04–22.80	0.001
*TP53 **	2	0	0	0	0	0	0	0	0	2	0	10.91	1.27–158.2	0.0647
*PALB2*	2	3	0	0	3	3	2	0	0	13	3	6.89	2.04–22.80	0.001
*NBN*	1	1	0	0	0	0	0	0	0	2	2	1.57	0.25–10.06	0.6448
*CDH1*	0	0	0	1	0	0	0	0	0	1	0	n/a		
*RAD51C*	0	0	0	0	0	0	0	0	0	0	0	n/a		
*RAD51D*	0	0	0	0	0	0	0	0	0	0	0	n/a		
*BRIP1*	1	0	0	2	0	2	0	0	0	5	2	3.95	0.85–19.9	0.1172
*BARD1*	1	0	0	0	0	0	0	0	0	1	1	1.57	0.08–29.9	>0.9999
*PTEN*	1	0	0	0	0	0	0	0	0	1	0	n/a		
Total PVs	32	7	1	10	19	15	15	18	9	126	24			
Total tested	487	66	63	53	108	59	37	33	12	918	1443			
% PVs	6.6%	10.6%	1.6%	18.9%	17.6%	25.4%	40.5%	54.5%	75.0%	13.7%	1.7%			
Total non-BRCA PVs	26	6	0	6	6	8	3	2	0	57				
% non-BRCA PVs	5.3%	9.1%	0.0%	11.3%	5.6%	13.6%	8.1%	6.1%	0.0%	6.2%				
% *BRCA1/2* PVs	1.2%	1.5%	1.6%	7.5%	12.0%	11.9%	32.4%	48.5%	75.0%	7.5%				
% adjusted for all *BRCA1/2* tested	5.3%	8.8%	0.0%	10.6%	5.0%	10.4%	5.5%	2.6%	0.0%	5.3%				
Negative pre-screen														
*ATM*	1	0	1	0	4	2	2	2	0	12	5	7.4	2.7–19.0	<0.0001
*CHEK2*	0	0	1	2	4	2	0	2	1	12	3	12.3	3.5–41.0	<0.0001
*TP53 **	1	0	0	0	4	1	1	0	1	8	0	84.7	13.2–941.5	<0.0001
*PALB2*	0	0	3	1	3	3	1	0	0	11	3	11.3	3.5–37.9	<0.0001
*NBN*	0	0	0	0	1	0	0	0	0	1	2	1.50	0.1–12.9	0.5777
*CDH1*	0	0	0	0	0	0	0	0	0	0	0	n/a		
*RAD51C*	0	0	0	0	0	0	1	0	0	1	0	n/a		
*RAD51D*	0	0	0	0	0	0	1	0	0	1	0	n/a		
*BRIP1*	0	0	0	0	1	1	0	0	0	2	2	3.02	0.5–19.3	0.261
*BARD1*	0	0	0	0	0	0	0	0	0	0	1	n/a		
*PTEN*	0	0	0	0	0	0	0	0	0	0	0	n/a		
Total PVs	2	0	5	3	17	9	6	4	2	48	16			
Total tested	38	12	28	53	147	90	58	39	15	480	1443			
% PVs	5.3%	0.0%	17.9%	5.7%	11.6%	10.0%	10.3%	10.3%	13.3%	10.0%	1.1%			
% adjusted for all *BRCA1/2* tested	5.2%	0.0%	17.1%	5.3%	10.3%	7.7%	7.0%	4.5%	2.8%	8.5%				
All samples tested for *BRCA1/2*														
*BRCA1*	4	5	5	16	49	87	60	94	67	387				
*BRCA2*	9	7	15	22	82	86	66	65	25	377				
Total tested	878	326	468	619	1240	745	387	281	116	5060				
% *BRCA1/2* PVs	1.5%	3.7%	4.3%	6.1%	10.6%	23.2%	32.6%	56.6%	79.3%	15.1%				

* Population frequency estimate for local population taken as 1 in 5000; NA-Not assessable as 480 pre-screened for *BRCA1/2*.

**Table 2 cancers-13-04154-t002:** Rate of PVs by pathology and Manchester score.

	All Patients Tested for BRCA1/2 PVs (*n* = 5060)						Patients Included for Panel Testing (*n* = 1398)				
	*BRCA1*	%	*BRCA2*	%	Total	Total PVs *BRCA1/2*	Panel Positive	%	Total Panel Tested	Non BRCA	Genes
Lobular total	4	1.2%	19	5.6%	341	6.74%	11	8.6%	128	7.8%	ATM(4), BRCA2(1), CDH1(1), CHEK2 (1), TP53(1), PALB2(3)
Lobular MSS ≥ 20	2	2.7%	8	10.7%	75	13.3%	5	11.9%	42	11.9%	ATM(2), TP53 (1), PALB2(2)
Lobular MSS < 20	2	0.75%	11	4.1%	266	4.9%	6	6.98%	86	5.8%	ATM(2), BRCA2(1), CDH1(1), CHEK2(1), PALB2(1)
Grade 1 total	3	0.8%	12	3.3%	363	4.1%	8	6.1%	132	4.55%	ATM(2), BRCA2(2), BRIP1(2), CHEK2(1), TP53(1)
Grade 1 MSS ≥ 20	3	4.8%	3	4.7%	63	9.5%	1	6.7%	15	6.7%	BRIP1(1)
Grade 1 MSS < 20	0	0.0%	9	3.0%	300	3.0%	7	5.98%	117	4.3%	ATM(2), BRCA2(2), BRIP1(1), CHEK2(1), TP53(1)
IDC Grade 2 ER+ total	10	1.3%	70	9.2%	763	10.5%	23	8.98%	256	4.3%	ATM(6), BRCA2(13), BRIP1(1), CHEK2(2), PALB2(1), RAD51C(1)
IDC Grade 2 ER+ MSS ≥ 20	8	4.3%	38	20.5%	185	24.8%	13	25.5%	51	7.8%	ATM(2), BRCA2(9), PALB2(1), RAD51C(1)
IDC Grade 2 ER+ MSS < 20	2	0.35%	32	5.5%	578	5.9%	10	4.9%	205	3.4%	ATM(4), BRCA2(3), BRIP1(1), CHEK2(2)
IDC Grade 3 ER+ total	43	8.4%	74	14.4%	513	22.8%	28	18.7%	150	10.0%	ATM(4), BRCA1(6), BRCA1&BRCA2(1), BRCA2(6), CHEK2(3), PALB2(5), RAD51D(1), TP53(2)
IDC Grade 3 ER+ MSS ≥ 20	34	17.3%	54	27.4%	197	44.7%	17	38.6%	44	15.9%	ATM(2), BRCA1(6), BRCA1&BRCA2(1), BRCA2(3),CHEK2(1), PALB2(2), RAD51D(1), TP53(1)
IDC Grade 3 ER+ MSS < 20	9	2.85%	20	6.33%	316	9.2%	11	10.4%	106	7.55%	ATM(2), BRCA2(3), CHEK2(2), PALB2(3), TP53(1)
TNT total	199	20.35%	61	6.24%	978	26.6%	31	17.5%	177	7.3%	ATM(1), BRCA1(14), BRCA2(4), BARD1(1), BRIP1(1), CHEK2(2), PALB2(7), RECQL(1)
TNT MSS ≥ 20	158	33.40%	46	9.73%	473	43.1%	19	24.4%	78	6.4%	BRCA1(12), BRCA2(2), BRIP1(1), CHEK2(1), PALB2(3)
TNT MSS < 20	41	8.12%	15	2.97%	505	11.1%	12	12.1%	99	8.1%	ATM(1), BRCA1(2), BRCA2(2), BARD1(1), CHEK2(1), PALB2(4), RECQL(1)
HER2+ total	5	1.72%	13	4.5%	290	6.2%	10	9.8%	102	7.8%	ATM(3), BRCA2(2), CHEK2(1), PALB2(1), TP53(3)
HER2+ MSS ≥ 20	2	4.88%	3	7.3%	41	12.2%	1	8.3%	12	8.3%	PALB2(1)
HER2+ MSS < 20	3	1.20%	10	4.2%	249	5.2%	9	10.0%	90	7.8%	ATM(3), BRCA2(2), CHEK2(1), TP53(3)
DCIS total	8	2.80%	22	7.7%	286	10.5%	18	15.25%	118	9.3%	ATM(4), BRCA2(7), BRIP1(1), CHEK2(3), PALB2(2), TP53(1)
DCIS MSS ≥ 20	4	7.55%	11	20.75%	53	28.3%	5	33.3%	15	13.3%	ATM(1), BRCA2(3), TP53(1)
DCIS MSS < 20	4	1.72%	11	4.7%	233	6.4%	13	12.6%	103	8.7%	ATM(3), BRCA2(4), BRIP1(1), CHEK2(3), PALB2(2)
NOS total	115	7.54%	106	6.95%	1526	14.5%	41	12.2%	335	8.4%	ATM(7), BRCA1(5), BRCA2(8), BRIP1(1), CHEK2(12), PALB2(5), PTEN(1), TP53(2)
NOS MSS ≥ 20	97	21.95%	79	17.9%	442	39.8%	16	18.6%	86	10.5%	ATM(4), BRCA1(4), BRCA2(3), BRIP1(1), CHEK2(4)
NOS MSS < 20	18	1.66%	27	2.5%	1084	4.15%	25	10.0%	249	7.6%	ATM(3), BRCA1(1), BRCA2(5), CHEK2(8), PALB2(5), PTEN(1), TP53(2)
Total overall	387	7.65%	377	7.45%	5060	15.1%	170	12.2%	1398	7.3%	ATM(31), BRCA1(25), BRCA2(42),BRCA1&BRCA2(1), BARD1(1), BRIP1(6), CDH1(1), CHEK2(25), PALB2(24), PTEN(1), TP53(10), RAD51C (1), RAD51D(1), RECQL(1)
MSS ≥ 20	308	20.14%	242	15.8%	1529	35.9%	77	22.45%	343	9.9%	ATM (11), BRCA1(22), BRCA2(20), BRCA1&BRCA2(1), BRIP1(3), CHEK2(6), PALB2(9), TP53(3), RAD51C (1), RAD51D(1)
MSS < 20	79	2.24%	135	3.8%	3531	6.1%	93	8.8%	1055	6.4%	ATM (20), BRCA1(3), BRCA2(22), BARD1(1), BRIP1(3), CDH1(1), CHEK2(19), PALB2(15), PTEN(1), TP53(7), RECQL(1)

## Data Availability

Data is available on request.

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
