# Peer review of "Gene Panel Testing for Breast Cancer Reveals Differential Effect of Prior BRCA1/2 Probability"

_cancers, 2021, doi:10.3390/cancers13164154_

Round 1

Reviewer 1 Report

Cancers

Article

 Gene panel testing for breast cancer reveals differential effect 2 of prior BRCA1/2 probability written by Dr D. Gareth Evans et al.

 Simple Summary: Previous studies reporting large gene panels in breast cancer have mainly reported on the contribution of additional genes compared to BRCA1/2. We have shown a very large differential additional effect from non-BRCA genes dependent on a priori likelihood of BRCA1 and BRCA2 combined. We have undertaken panel testing beyond BRCA1/2 in 1398 patients with breast cancer and identified 95 (6.3%) with actionable breast cancer genes. The highest rate was found for grade 3 ER+ Her2- breast cancers. Those with the lowest likelihood of BRCA1/2 by Manchester score had a 3-fold higher rate of non-BRCA genes whereas those with predicted rats of ~80% had 5-fold higher rate of BRCA1/2. Unless those referring patients with breast cancer for extended panel testing are certain there is no loss of sensitivity for BRCA1/2 they should opt of a bespoke BRCA1/2 test first in those with high prior likelihoods of BRCA1/2 pathogenic variants.

  This reviewer thinks the manuscript “Gene panel testing for breast cancer reveals differential effect 2 of prior BRCA1/2 probability” written by Dr D. Gareth Evans et al. is clinically important, interesting and informative, however, some additional description will be required to clarify the concept.

  1. Did the pathogenic variants (PVs) of gene panel testing for breast cancer genes (ATM,BARD1,CDH1,CHEK2,PALB2,PTEN,RAD51C, 34 RAD51D,TP53) returned to the patients or not? Are the patients all females?
  2. How did perform the genetic counselling or informed consent to the study participants? Please describe the ethical issues and annotation procedures of PVs in the materials and methods.
  3. How did the annotation of PVs obtain? What kind of database was used?
  4. Saliva and blood samples were used in this study. What is the difference of the sample selection?
  5. Please describe the result of the PVs of MMR (MLH1, MSH2, MSH6 and PMS2) genes in the table and how do the authors interpret the results.
  6. In conclusion, “The majority of the increased diagnostic yield results from just three genes PALB2, CHEK2 and Only PALB2 can be considered a strong enough risk factor on its own and the actionability of ATM and CHEK2 pathogenic variants needs to be taken in the context of other risk factors”. Some report shows RAD51C, RAD51D, BRIP1, or BARD1 contributes to increase the probability of HBOC. Do the authors think that PVs of RAD51C, RAD51D, BRIP1, and BARD1 are not related to grade 3 ER+ Her2- breast cancers besides HBOC. The authors need to describe the percentage of ovarian cancer complication of the patients.

7. In case PVs were detected in breast cancer patients participated in this study, the genetic counselling or the confirmation gene test was performed or not? 

Author Response

Reviewer 1

This reviewer thinks the manuscript “Gene panel testing for breast cancer reveals differential effect 2 of prior BRCA1/2 probability” written by Dr D. Gareth Evans et al. is clinically important, interesting and informative, however, some additional description will be required to clarify the concept.

 >Thank you

  1. Did the pathogenic variants (PVs) of gene panel testing for breast cancer genes (ATM,BARD1,CDH1,CHEK2,PALB2,PTEN,RAD51C, 34 RAD51D,TP53) returned to the patients or not? Are the patients all females?

>Thank you. All results from MCGM clinical panel testing have been returned. We are awaiting ethical approval to approach PROCAS participants to see if they wish to receive their results. There were no males included. We have clarified this in first line of methods ‘A total of 5060 women with breast cancer and 1443 control samples were included in the study. Male cases were excluded.’ We have also added ‘PROCAS women only initially provided consent for return of a polygenic risk score but also consented to further research genetic testing. We are applying for ethical approval to approach women with panel testing to assess whether they wish to receive results.’

  1. How did perform the genetic counselling or informed consent to the study participants? Please describe the ethical issues and annotation procedures of PVs in the materials and methods.

>We have clarified consent issues above. We have added ‘The samples clinically tested at MCGM were annotated by a clinically approved national laboratory hub. Annotation of research results from BRIDGES were initially confirmed by the BRIDGES team and confirmed by our in-house research group (EvV).’ To the methods section.

  1. How did the annotation of PVs obtain? What kind of database was used?

> Pathogenic variants were annotated according to ACMG guidelines. We have separated out the Methods sections into “Study participants”, “Genetic testing” and “Calculated likelihood of a BRCA1/2 pathogenic variant” for clarity and have now referred to this in the “Genetic testing” section.

  1. Saliva and blood samples were used in this study. What is the difference of the sample selection?

> The source of DNA is different depending on whether the patients were recruited through the MCGM (blood samples) or PROCAS (saliva), but either can be used for genetic testing. We have indicated this in the “Study population” section of the Methods

  1. Please describe the result of the PVs of MMR (MLH1, MSH2, MSH6 and PMS2) genes in the table and how do the authors interpret the results.

>Thank you. We have added a supplementary table with the only MMR gene that was identified in cases. As MSH6 was not tested in the whole dataset who do not feel it should go in the main table as it is not currently approved as an  actionable gene by NCCN and the findings of various studies are contradictory as we state in the discussion. We do not consider the evidence yet compelling enough for MSH6 and that the other MMR genes are not to be considered as breast cancer genes.

  1. In conclusion, “The majority of the increased diagnostic yield results from just three genes PALB2, CHEK2 and Only PALB2 can be considered a strong enough risk factor on its own and the actionability of ATM and CHEK2 pathogenic variants needs to be taken in the context of other risk factors”. Some report shows RAD51C, RAD51D, BRIP1, or BARD1 contributes to increase the probability of HBOC. Do the authors think that PVs of RAD51C, RAD51D, BRIP1, and BARD1 are not related to grade 3 ER+ Her2- breast cancers besides HBOC. The authors need to describe the percentage of ovarian cancer complication of the patients.

>Thank you. We have added details of ovarian cancer in supplementary tables 2 and 3. The following text has been added to the text ‘A total of 134 women with breast cancer had either a personal (n=25) or family his-tory (n=109) of ovarian cancer. Detection rates were similar to those with breast cancer only with 8/134 (6%) having a PV beyond BRCA1/2 (PALB2=2) and 2/25 (8%) with breast and ovarian double primaries (Supplementary tables 2 and 3).’ We agree that RAD51C, RAD51D and BRIP1 are actionable ovarian cancer genes and that RAD51C, RAD51D, and BARD1 are now established as predisposing to triple negative breast cancer. However. Our study was not large enough to confirm these findings. Nonetheless, 2 PALB2 PVs were found in 134 women with a personal or family history of ovarian cancer.

  1. In case PVs were detected in breast cancer patients participated in this study, the genetic counselling or the confirmation gene test was performed or not?

> PVs detected by multigene panel were verified by Sanger sequencing. We have now indicated this in the “Genetic testing” section of the Methods.

Reviewer 2 Report

The proposed paper is readily understandable because it is well-constructed, clear and well described with figures exhaustive and appropriate to the subject matter. The scientific background, aims are clearly explained and seem to be appropriate to the investigation field. The conclusions logically follow from the data reported in literature. The information collected in this paper could have important implications for a better understanding of the tumor biology and for identifying new molecular mechanisms underlying cancer. The topic falls within the scope of the subject area of the journal.

In my opinion, this work is acceptable for publication in Cancers after minor revisions which will help the authors to improve the quality of their manuscript.

  • In the paragraph of ‘materials and methods’ insert a 'study population' paragraph and clarify the enrolment of patients;
  • Please specify which criteria you have used to select patients for multigenic panel testing.
  • In the Discussione, please cite the following article (https://doi.org/10.3390/cancers12092415).

Author Response

Reviewer 2

The proposed paper is readily understandable because it is well-constructed, clear and well described with figures exhaustive and appropriate to the subject matter. The scientific background, aims are clearly explained and seem to be appropriate to the investigation field. The conclusions logically follow from the data reported in literature. The information collected in this paper could have important implications for a better understanding of the tumor biology and for identifying new molecular mechanisms underlying cancer. The topic falls within the scope of the subject area of the journal.

>Thank you

In my opinion, this work is acceptable for publication in Cancers after minor revisions which will help the authors to improve the quality of their manuscript.

  • In the paragraph of ‘materials and methods’ insert a 'study population' paragraph and clarify the enrolment of patients;
  • Please specify which criteria you have used to select patients for multigenic panel testing.
  • In the Discussione, please cite the following article (https://doi.org/10.3390/cancers12092415).

> Thank you. We have now re-formatted the Methods section to separate out the study population and the methodology, and indicating which patients were included in the multigene panel testing. PROCAS cases were selected as a case control series for BRIDGES. There was some enrichment of testing for those with high Manchester scores to try to identify the missing heritability. We have also added the recommended reference to the manuscript, but felt it was more appropriate in the introduction than the Discussion.

Round 2

Reviewer 1 Report

The authors have responded all the inquiries and described properly in the manuscript by this reviewer. This reviewer approved all the comments for the revised version of the manuscript entitled “Gene panel testing for breast cancer reveals differential effect of prior BRCA1/2 probability” by Dr D. Gareth Evans et al, is ready for publication consideration in the editorial office of “Cancers.